# Local Melon and Watermelon Crop Populations to Moderate Yield Responses to Climate Change in North Africa

Stuart Alan Walters [1,*], Mimouni Abdelaziz [2] and Rachid Bouharroud [2]

1    School of Agricultural Sciences, Southern Illinois University, Carbondale, IL 62901, USA
2    Integrated Crop Production Unit, Regional Center of Agadir, National Institute of Agronomic Research, Agadir 80350, Morocco; abdelaziz.mimouni@inra.ma (M.A.); rachid.bouharroud@inra.ma (R.B.)
*    Correspondence: awalters@siu.edu; Tel.: +1-618-4532496

**Abstract:** Climate change is having a tremendous influence on world food production, with arid, semi-arid, and dry sub-humid areas especially susceptible. In these areas, locally adapted crop varieties or landraces can be used to mitigate the influence of climate change on current and future food security challenges. The high genetic diversity within these populations allows for crops to adapt to changing environments or other stresses that influence growth and productivity. Thus, local Moroccan melon (*Cucumis melo*) and watermelon (*Citrullus lanatus*) landraces were compared to pure-line varieties in southwestern Morocco to identify their adaptability and possible ability to mitigate current and future climate change. Results indicated that the melon and watermelon landraces evaluated most likely could help mitigate yield losses from climate change in this area of Morocco. 'AitOulyad', a local muskmelon type, and 'Rasmouka Ananas' were both outstanding melon landraces with high plant vigor and yields. For watermelon, 'AitOulyad' had extremely high yields but had high numbers of seed in the flesh, while 'Rasmouka' had a lower yield, fewer seeds in the flesh, and a higher fruit consistency. This research indicates that melon and watermelon landraces in this area of southwestern Morocco with a semi-arid to arid climate will continue to play a major role in crop adaptation to maintain high productivity under a rapidly changing environment.

**Keywords:** climate change; crop population; genetic resources; genetic variability; vegetable landrace

## 1. Introduction

In vegetable production systems throughout the world, climate change is having tremendous influences on food production. Climate change is directly or indirectly influencing many factors of vegetable crop production including water availability, temperature extremes during production cycles, soil fertility, and pest populations [1–3]. However, climate change is having more drastic influences on the effectiveness of vegetable production in developing countries that experience extremes in heat and rainfall patterns. Specific climatic conditions, such as changing rainfall patterns and humidity, light intensity, temperature, solar radiation, and $CO_2$ concentration can all impact vegetable crop physiological responses which can have detrimental influences on resulting yields and product quality [4].

World food security is a challenge that must be met in part by mitigating plant responses to climate change, while at the same time, conserving essential natural resources so that effective food production activities can be sustained for generations [3,5]. The rapid ongoing change in the earth's climate is a major challenge to food production worldwide, especially those fringe food production regions that are barely able to support agricultural activities under normal conditions. Arid, semi-arid, and dry sub-humid areas are especially susceptible to the negative effects of climate change as many of these areas are often classified as fringe crop production areas anyway. These drylands encompass around 38% of the Earth's land area, covering much of North and Southern Africa, Western North America, Australia, the Middle East, and Central Asia, and are home to approximately

2.7 billion people of which about 90% of whom live in developing countries [6]. Water is an increasingly limited resource influenced by a changing climate and has a definite influence on the long-term productivity of world agriculture [7]. Crop plant adaptation to water shortages and optimizing water management in production systems are both important to alleviate climate changes in arid and semi-arid Africa.

Traditional agricultural production systems have played a vital role in the evolution and conservation of on-farm diversity, allowing farmers to circumvent crop failure by reducing vulnerability to environmental stresses [8]. The utilization of more locally adapted crop germplasm (e.g., landraces or local crop varieties) to mitigate the effects of drought due to fluctuating water supplies is a strategy that can be used to cope with these ongoing and future food security challenges [7]. Crop landraces (or local crop varieties) are highly heterogeneous mixtures with slightly differing genotypes that generally produce plant phenotypes having small but often important differences under field conditions [9]. They have high amounts of genetic diversity, but often have lower quality and yields with inconsistent phenotypes compared to newer hybrids and/or even older pure-line inbred varieties [10]. Additionally, due to the high costs associated with hybrid vegetable seed, most small growers in developing countries have chosen the safest route possible for revenue generation, which is to use either landraces or pure line inbred varieties. Although landraces typically have little cost, pure line varieties generally provide modest yields with consistent appearance and quality characters at a low cost compared to hybrids [10].

Humans over millennia have inadvertently influenced crop evolution by saving and planting seeds from crop plants with the most desirable characteristics, which has allowed the selection of adapted genotypes that perform best for specific climatic conditions [9]. Vegetable crops grown in sustainable, low-input, traditional farming systems have generally been landraces. Today, landraces are still grown in many developing countries, although their cultivation is diminishing [10]. Landraces are highly heterogeneous mixtures of crop population genotypes with individuals often providing differing plant phenotypes and resulting yields. The genetic diversity contained within landrace populations is an important part of global crop diversity and is considered of paramount importance for future world crop production [11]. However, the displacement of locally adapted landraces with improved hybrids and cultivars is a threat to biodiversity and potential adaptation to future climate changes [12]. Walters et al. [7] indicated that in the Souss-Massa region of southwestern Morocco, 31% of small, subsistence farmers cultivated vegetable landraces and saved seed, with two highly important crops in this region being melon (*Cucumis melo*) and watermelon (*Cucumis lanatus*). Watermelon and melon landraces are often cultivated in remote arid to semi-arid regions of the developing world that are most vulnerable to climate change. In Morocco, melon and watermelon are grown in most parts of the center and south of the country with a concentration in the Marrakech and Souss-Massa regions [13].

There is little information available regarding melon and watermelon landraces grown in the semi-arid to arid regions of North Africa, although Walters et al. [7] indicated that Morocco's vegetable crop genetic diversity was diminishing due to the loss of landraces being cultivated. In the harsh hot and dry, arid to the semi-arid climate of the Souss-Massa region in southwestern Morocco, vegetable crops are grown under significant heat and drought stress conditions which can limit their productivity and quality [7]. These stress conditions can be exacerbated under climate changes and adapted landrace genotypes have generally performed well in these situations. Therefore, the purpose of this study was to evaluate local Moroccan landraces of these two cucurbits with other pure line varieties that are available in local markets to observe their growth and productivity under this harsh semi-arid climate of southwestern Morocco. The goal of this research was to identify those melon or watermelon landraces that could possibly be used to ease the influences of current and future climate changes.

## 2. Materials and Methods

### 2.1. Study Location

Field experiments were conducted at the Moroccan National Institute for Agricultural Research farm located near Belfaa, Agadir, in southwestern Morocco (30°02′39.349″ N, 9°33′13.513″ W, Alt. 100 m). The soil at the research farm was a sandy limestone soil, with a very low organic matter content of <0.5% and a pH of 7.5 to 8.0.

### 2.2. Experimental Setup

Two studies (melon and watermelon) were conducted during both the 2015 and 2016 growing seasons. The melon experiment evaluated five Moroccan landraces and one pure line variety as the control, while for watermelon, two Moroccan landraces and three pure lines varieties were evaluated. The melon landraces were named based on the location in which they were collected from local melon growers: 'Ait Baha', 'AitOulyad', and 'Casablanca' are local melons, which are large muskmelon types; 'Rasmouka' ananas is an oval, netted melon with a pineapple aroma and white to yellow flesh, and 'Marrakech' souihla is a round netted melon with green flesh. 'Ait Baha' (30°4′33.5″ N, 9°9′26.7″ W), 'AitOulyad' (30°14′24.7″ N, 9°22′28″ W), 'Rasmouka' ananas (29°48′1.6″ N, 9°32′58.9″ W), and 'Marrakech' souihla (31°38′17.2″ N, 7°58′51.9″ W) were collected from locations in the southwestern area of Morocco, while 'Casablanca' muskmelon (33°35′48.4″ N, 7°36′35.3″ W) was collected from an agricultural store in Casablanca, which is near the central coastal region of Morocco. They had obtained this landrace seed from growers producing melons close to Casablanca in a cooler and more humid climate. 'Charentais' is widely grown in the region and was included as the pure-line variety; it produces small fruit with a light gray to bluish skin color with firm orange flesh that has a distinct and intense aroma. The watermelon experiment included three pure line varieties ('Orangeglo', 'Bush Sugar Baby' and All Sweet') that some small landowners in arid to semi-arid regions of developing countries are currently growing, and two landraces, 'AitOulyad' and 'Rasmouka' from the region. These were again named for the locations in which they were collected in southwestern Morocco. For both the melon and watermelon experiments, a randomized complete block design was used with five replications.

Melon and watermelon field experiments were planted mid-April in 2015 and 2016 to provide the greatest heat stress conditions on these plants, as summer temperatures can often reach 48 °C in July and August at this location. Organic sheep manure was broadcast applied at 2 metric ton/ha and tilled into the soil, prior to bed formation. Plants were grown on 12 cm raised beds, that were 2.1 m center-to-center, and covered with 1.25 mil black plastic with drip irrigation underneath. Melon and watermelon plants were watered twice weekly with a total of an additional 75 kg N per ha fertigated over the growing season. Experimental units contained five plants with each watermelon plant spaced 1.2 m apart and melon plants spaced 0.6 m apart. Although weeds were controlled manually by hand, standard insecticides (abamectin, deltamethrin, spiromesifen) and fungicides (hexaconozol, mancozeb, myclobutanil) were effectively rotated using four applications in June to manage insect and disease pests.

### 2.3. Data Collection

Various data were collected to determine the suitability of both melon and watermelon landraces/pure line varieties to the harsh Moroccan climate. A foliar disease rating (0 = none, 1–3 = low disease, 4–5 = moderate disease, and 7–9 = high disease) was collected at ~80 days after seeding. The prevalent diseases were *Fusarium* wilt in watermelon, caused by *Fusarium oxysporum f.* sp. *niveum* (FON), and Gummy stem blight and powdery mildew in melons, caused by *Didymella bryoniae* and *Podosphaera xanthii*, respectively. Plant vigor was rated as 1 to 3 = highly vigorous plants and healthy foliage, 4 to 6 = moderate vigor with foliage lighter green in color, and 7–9 = weak plants with low vigor and yellowing foliage. Melon and watermelon fruit yield (numbers and weights) were collected at several harvests, as the fruit ripened. The numbers of seeds per fruit were also enumerated as small

growers prefer varieties/landraces that produce high amounts of seed for the following years' plantings. A fruit consistency rating from 0 to 100% (based on all fruit harvested) was also collected from each experimental unit and was used to determine the consistency of fruit size and shape.

*2.4. Data Analysis*

Data were subjected to analysis of variance procedures appropriate for a randomized complete block design using SAS (SAS Inst., Cary, NC, USA). Fisher's least significant difference (LSD) tests were used to separate differences among means at $p \leq 0.05$ for the variables collected for both melon and watermelon.

**3. Results**

The data were combined over the 2 growing seasons and analyzed. For both the melon and watermelon experiments, no interactions were observed ($p > 0.05$) between years and the landraces/pure line varieties evaluated for the variables collected. Therefore, results for all variables were combined over the two years.

*3.1. Melon Evaluation*

The melon landraces evaluated all provided differing characteristics, regarding foliar disease susceptibility, plant vigor, consistency of fruit shape and size, yield, average fruit weight, and seed per fruit (Table 1). This was expected since melon landraces grown closer to where they were collected should provide better results. Those landraces that were close to the evaluation site conducted at the INRA research experimental farm in Belfaa were 'AitOulyad' and 'Rasmouka' ananas, followed by 'Ait Baha' and 'Marrakech' souihla, with 'Casablanca' landrace being the farthest away.

**Table 1.** Melon (*Cucumis melo*) landrace/pure line evaluations in Belfaa, Morocco combined for 2015 and 2016.

| Melon Variety/ Landrace | Type [a] | Foliar Disease Rating [b] | Plant Vigor Rating [c] | Fruit Consistency (0–100%) [d] | Fruit No. per ha | Fruit wt (kg) per ha | Average Fruit wt (kg) | Seed No. per Fruit |
|---|---|---|---|---|---|---|---|---|
| AitOulyad muskmelon | L | 7.4 a | 8.5 a | 95 a | 34,830 b | 111,897 a | 3.2 a | 2170 a |
| Ait Baha muskmelon | L | 7.1 a | 8.7 a | 95 a | 29,622 c | 50,357 b | 1.7 b | 1878 a |
| Casablanca muskmelon | L | 4.5 b | 6.1 c | 90 a | 26,898 c | 40,347 b | 1.5 b | 996 c |
| Rasmouka ananas | L | 3.0 c | 7.1 b | 70 c | 42,141 a | 105,352 a | 2.5 ab | 1391 b |
| Marrakech souihla | L | 3.0 c | 7.9 ab | 80 b | 35,864 b | 43,037 b | 1.2 b | 832 c |
| Charentais | PL | 3.0 c | 6.0 c | 90 a | 42,269 a | 50,723 b | 1.2 b | 716 c |

[a] Type: L = landrace, PL = pure line. [b] Foliar disease rated as: 0 = none, 1–3 = low disease, 4–5 = moderate disease, and 7–9 = high disease. [c] Plant vigor was rated as: 1 to 3 = highly vigorous plants and healthy foliage, 4 to 6 = moderate vigor with foliage lighter green in color, and 7–9 = weak plants with low vigor and yellowing foliage. [d] Fruit Consistency rated from 0 to 100%, based on consistency of fruit size and shape. Means within the same column followed by the same letter are not significantly different according to Fisher's protected LSD at $p \leq 0.05$.

3.1.1. Foliar Disease Development

Foliar diseases are a major production issue in melon production in dry climates. Powdery mildew and gummy stem blight were both detected at a high incidence and provided significant foliar disease. Two landraces were highly susceptible to foliar disease development. 'AitOulyad' and 'Ait Baha' exhibited significant foliar disease compared to the other three landraces evaluated (Table 1; Figure 1). 'Casablanca', 'Rasmouka' ananas, and 'Marrakech' souihla exhibited less foliar disease over both years. 'Charentais' had a low foliar disease rating, with powdery mildew most often observed.

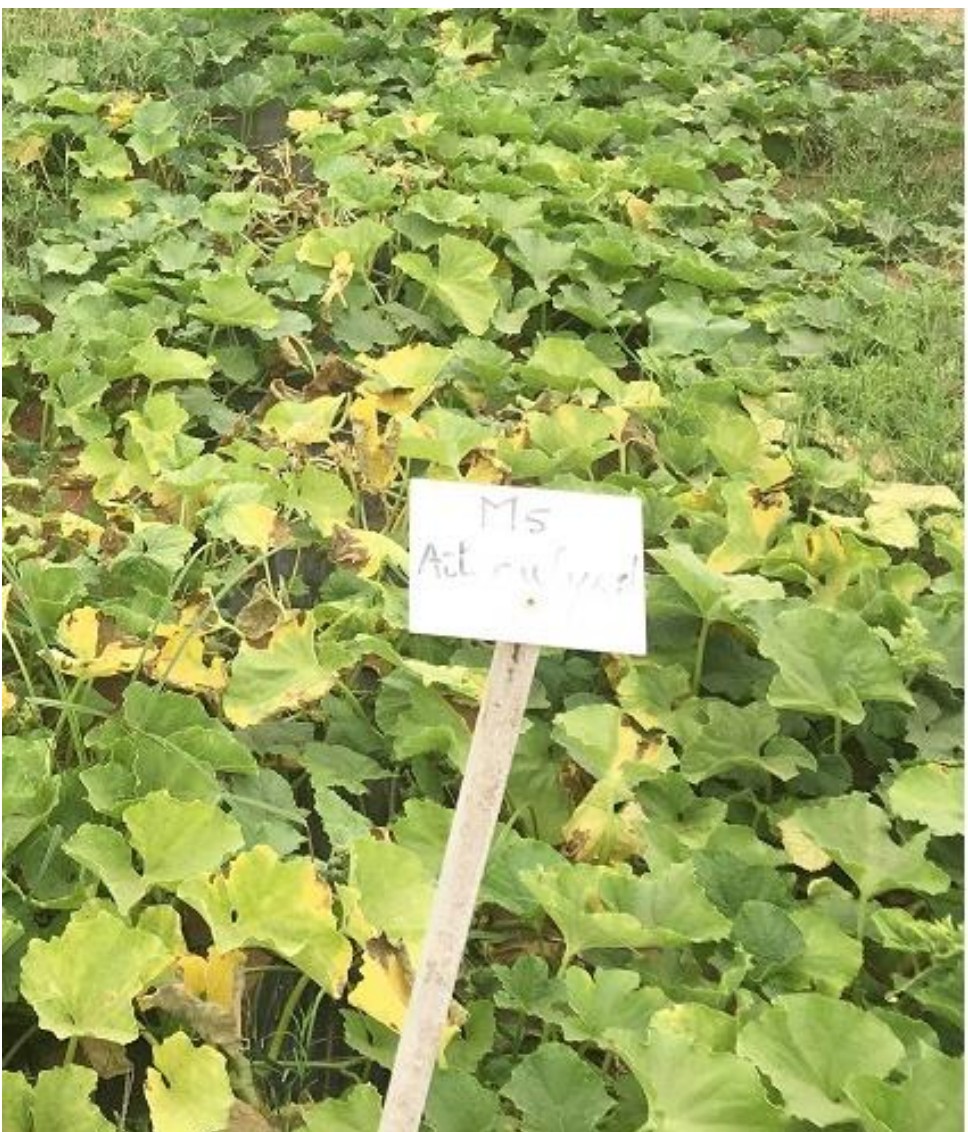

**Figure 1.** 'AitOulyad' muskmelon landrace showing high amounts of Gummy stem blight disease, caused by *Didymella bryoniae*.

### 3.1.2. Plant Vigor

The melon landraces differed ($p \leq 0.05$) for overall plant vigor (Table 1). 'AitOulyad', 'Ait Baha', and 'Marrakech Souihla' provided the greatest amount of plant vigor under the hot, dry growing conditions each year. 'Rasmouka Ananas' also provided high vigor although less than those three landraces previously described. 'Casablanca' had the lowest plant vigor of all melon landraces. 'Charentais' had a low plant vigor rating, having low vine growth compared to most landraces evaluated, except for 'Casablanca'.

### 3.1.3. Consistency of Melon Fruit Size and Shape

High amounts of fruit heterogeneity are often observed in melon landraces sold at local markets in Morocco (Figure 2), and the melon landraces/pure line variety evaluated differed ($p \leq 0.05$) in their ability to produce consistent fruit size and shape. 'AitOulyad', 'Ait Baha', and 'Casablanca were all 'local' muskmelon types and provided high fruit consistency ($\geq$90%). 'Marrakech Souihla' and 'Rasmouka Ananas' had less fruit consistency at 80% and 70%, respectively, while 'Charentais' produced a high fruit consistency at 90%.

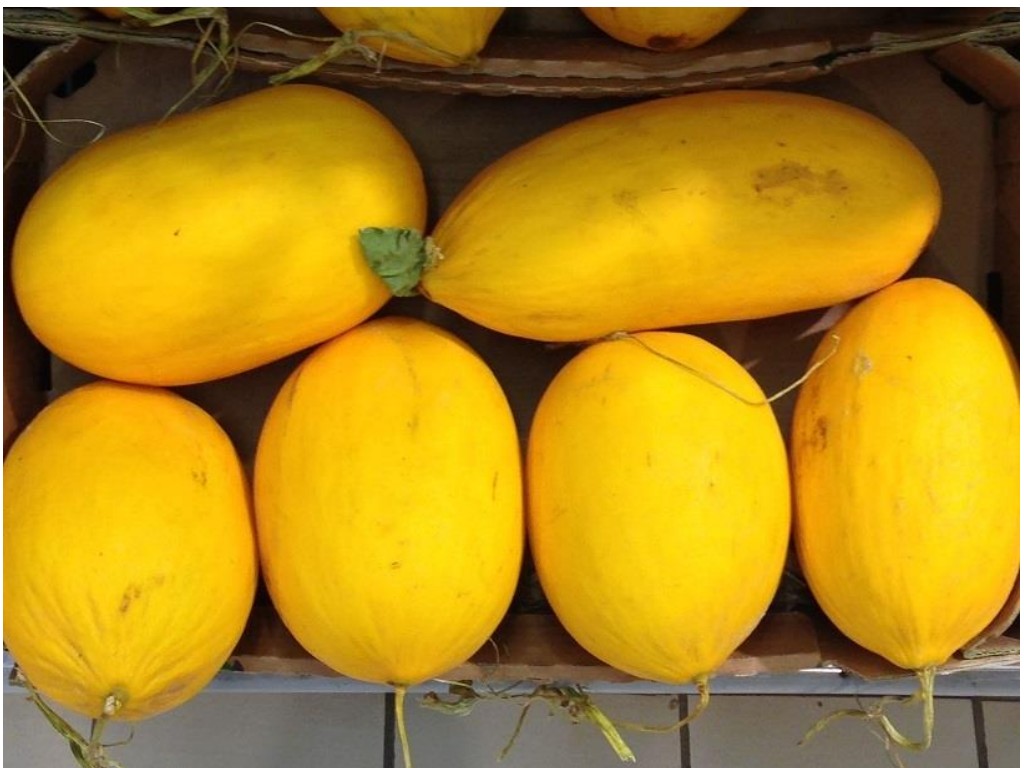

**Figure 2.** Canary melon landrace in Marrakech, Morocco local market. Notice fruit size and shape inconsistency.

### 3.1.4. Melon Fruit Number and Weight

Melon fruit yields differed ($p \leq 0.05$) among the landraces/pure line variety evaluated (Table 1). 'Rasmouka Ananas' produced high fruit numbers and weight per ha, with an average fruit size of 2.5 kg. 'AitOualyad' had the greatest average fruit size (3.2 kg) of all landraces evaluated. The other landraces had much lower fruit numbers and weights per ha and average fruit sizes, compared to 'AitOualyad' and 'Rasmouka Ananas', except for 'Marrakech Souihla' which produced similar fruit numbers per ha as 'AitOualyad'. 'Charentais' produced a high fruit number per ha similar to 'Rasmouka Ananas' but had low total fruit weights per ha and an average fruit weight of only 1.2 kg.

### 3.1.5. Seed Numbers per Melon

The melon landraces evaluated differed ($p \leq 0.05$) for the number of seeds in each fruit (Table 1). 'AitOualyad' and 'Ait Baha' produced the greatest amounts of seed per fruit followed by 'Rasmouka Ananas'. 'Casablanca' and 'Marrakech Souihla' had the least seed numbers per fruit for the landraces evaluated. 'Charentais' being a small-sized melon only produced only 716 seeds on average per fruit.

### 3.1.6. Summary for Melon Evaluation

This experiment indicated that melons differed for growth, disease susceptibility, and productivity. All local muskmelon types and the control 'Charentais' provided high fruit shape and size consistency, while the souihla and ananas types were less consistent. 'AitOulyad' muskmelon and 'Rasmouka Ananas' both provided high fruit numbers and weights per ha, with 'Charentais' and 'Marrakech Souilha' also producing high fruit numbers per ha. However, 'Casablanca' muskmelon provided much lower fruit weights per ha than 'AitOulyad' muskmelon and 'Rasmouka Ananas'. These four melon types ('Charentais', 'Marrakech Souilha', and 'Casablanca' and 'Ait Baha' muskmelon) also provided the lowest average fruits weight, with 'AitOulyad' muskmelon and 'Rasmouka Ananas' having much greater average melon size. 'Rasmouka Ananas' has an attractive

white to yellow flesh (Figure 3). 'AitOulyad' and 'Ait Baha' muskmelon produced the greatest numbers of seeds per fruit, followed by 'Rasmouka Ananas'. 'Casablanca' muskmelon and 'Marrakech Souilha' landraces provided low numbers of seed per fruit, with Charentais' producing the least. However, 'AitOulyad' and 'Ait Baha' muskmelon had significant foliar disease compared to the other three melon landraces evaluated. Foliage cover is an important characteristic to prevent sunburn of melon fruit in the southwestern Morocco climate, and 'Casablanca' muskmelon and 'Charentais' had poor plant vigor and provided less foliage cover for melon fruit, compared to the other melon landraces evaluated.

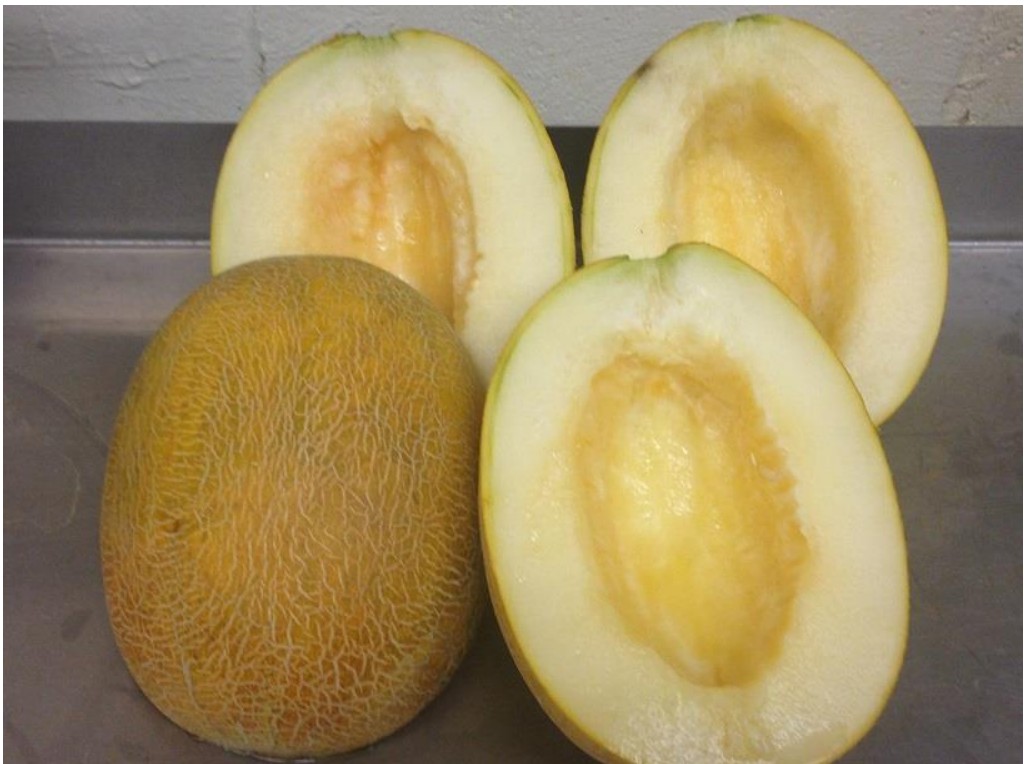

**Figure 3.** 'Rasmouka Ananas' cut fruit to show rind pattern and color, as well as its attractive interior fruit color.

*3.2. Watermelons*

The watermelon landraces/pure line varieties evaluated all provided differing characteristics, regarding foliar disease susceptibility, plant vigor, consistency of fruit shape and size, yield, average fruit weight, and seed per fruit (Table 2). Again, this was expected since watermelon landraces grown closer to where they were collected should provide better results. Both 'AitOulyad' and 'Rasmouka' were watermelon landraces that were collected within 50 km of the evaluation site at the INRA research farm in Belfaa. However, we searched extensively and only collected two watermelon landraces. They appear to be disappearing in the Souss region of Morocco and are not grown as frequently as melon landraces (7).

**Table 2.** Watermelon (*Citrullus lanatus*) variety/landrace evaluations in Belfaa, Morocco combined for 2015 and 2016.

| Watermelon Variety/Landrace | Type [a] | Foliar Disease Rating [b] | Plant Vigor Rating [c] | Fruit Consistency (0–100%) [d] | Fruits No. per ha | Fruit wt (kg) per ha | Average Fruit wt (kg) | Seed No. per Fruit |
|---|---|---|---|---|---|---|---|---|
| AitOulyad | L | 7.1 a | 5.3 b | 80 b | 11,208 a | 81,818 a | 7.3 c | 1179 a |
| All Sweet | PL | 3.1 c | 7.3 a | 90 ab | 8518 b | 78,366 a | 9.2 a | 490 c |
| Bush Sugar Baby | PL | 7.5 a | 2.4 d | 95 a | 4483 c | 18,380 c | 4.1 e | 744 b |
| Orangeglo | PL | 4.7 b | 6.8 a | 90 ab | 6725 b | 55,145 b | 8.2 b | 546 c |
| Rasmouka | L | 5.8 b | 4.1 c | 100 a | 7173 b | 45,907 b | 6.4 d | 739 b |

[a] Type: L = landrace, PL = pure line. [b] Foliar disease rated as: 0 = none, 1–3 = low disease, 4–5 = moderate disease, and 7–9 = high disease. [c] Plant vigor was rated as: 1 to 3 = highly vigorous plants and healthy foliage, 4 to 6 = moderate vigor with foliage lighter green in color, and 7–9 = weak plants with low vigor and yellowing foliage. [d] Fruit Consistency rated from 0 to 100%, based on consistency of fruit size and shape. Means within the same column followed by the same letter are not significantly different according to Fisher's protected LSD at $p \leq 0.05$.

### 3.2.1. Foliar Disease Development

The primary disease observed on watermelon was Fusarium wilt, and the watermelon landraces/pure line varieties differed for susceptibility to this disease (Table 2). 'AitOulyad' and 'Bush Sugar Baby' were both highly susceptible to this disease, followed by 'Rasmouka' and 'Orangeglo', while 'All Sweet' was the least affected by this disease.

### 3.2.2. Watermelon Plant Vigor

The watermelon landraces/pure line varieties evaluated differed ($p \leq 0.05$) for vigorous plant growth (Table 2). 'All Sweet' and 'Orangeglo' had the highest amount of plant vigor, followed by 'AitOulyad' and 'Rasmouka', with 'Bush Sugar Baby' being the least vigorous. Higher plant vigor was generally observed on those landraces/pure line varieties that had lower amounts of foliar disease (Table 2). 'Bush Sugar Baby' had both lower amounts of foliage compared to the other watermelon landraces/pure line varieties evaluated and was highly susceptible to Fusarium wilt.

### 3.2.3. Consistency of Watermelon Fruit Size and Shape

Differences ($p \leq 0.05$) were detected among the watermelon landraces/pure line varieties evaluated for fruit size and shape consistency. Most watermelons provided 90% or greater fruit consistency, except for 'AitOulyad' at 80%. High fruit consistency was expected in the pure line varieties, but 'Rasmouka' had the highest consistency for fruit size and shape, which was unusual since it is a landrace which should have some level of genetic heterogeneity.

### 3.2.4. Watermelon Fruit Number and Weight

Yield differences ($p \leq 0.05$) were observed among the watermelon landraces/pure line varieties evaluated. 'AitOulyad' and 'All Sweet' produced the greatest fruit weight per ha, followed by 'Orangeglo' and 'Rasmouka'. 'Bush Sugar Baby' provided only about 22 to 23% of the fruit weight per ha compared to 'AitOulyad' and 'All Sweet'. 'AitOulyad' fruit are slightly elongated with green striping on a light green background (Figure 4). Additionally, 'AitOulyad' produced the greatest fruit numbers per ha, followed by 'All Sweet', 'Rasmouka,' and 'Organeglo', with 'Bush Sugar Baby' producing the lowest. For average fruit weight, 'All Sweet' was the highest at 9.2 kg followed by 'Orangeglo at 8.2 kg, 'AitOulyad' at 7.3 kg, 'Rasmouka' at 6.4 kg, and 'Bush Sugar Baby' at 4.1 kg. 'All Sweet' watermelon provided low foliar disease, high plant vigor, good fruit consistency, high yields, and outstanding average fruit size, which made it the top choice in the evaluation, while 'Bush Sugar Baby' having poor foliar disease and low plant vigor, low yields and low average fruit size, was the least desirable. The two landraces evaluated were quite comparable although, 'AitOulyad' was probably slightly better than 'Rasmouka' due to it having a higher-yielding ability.

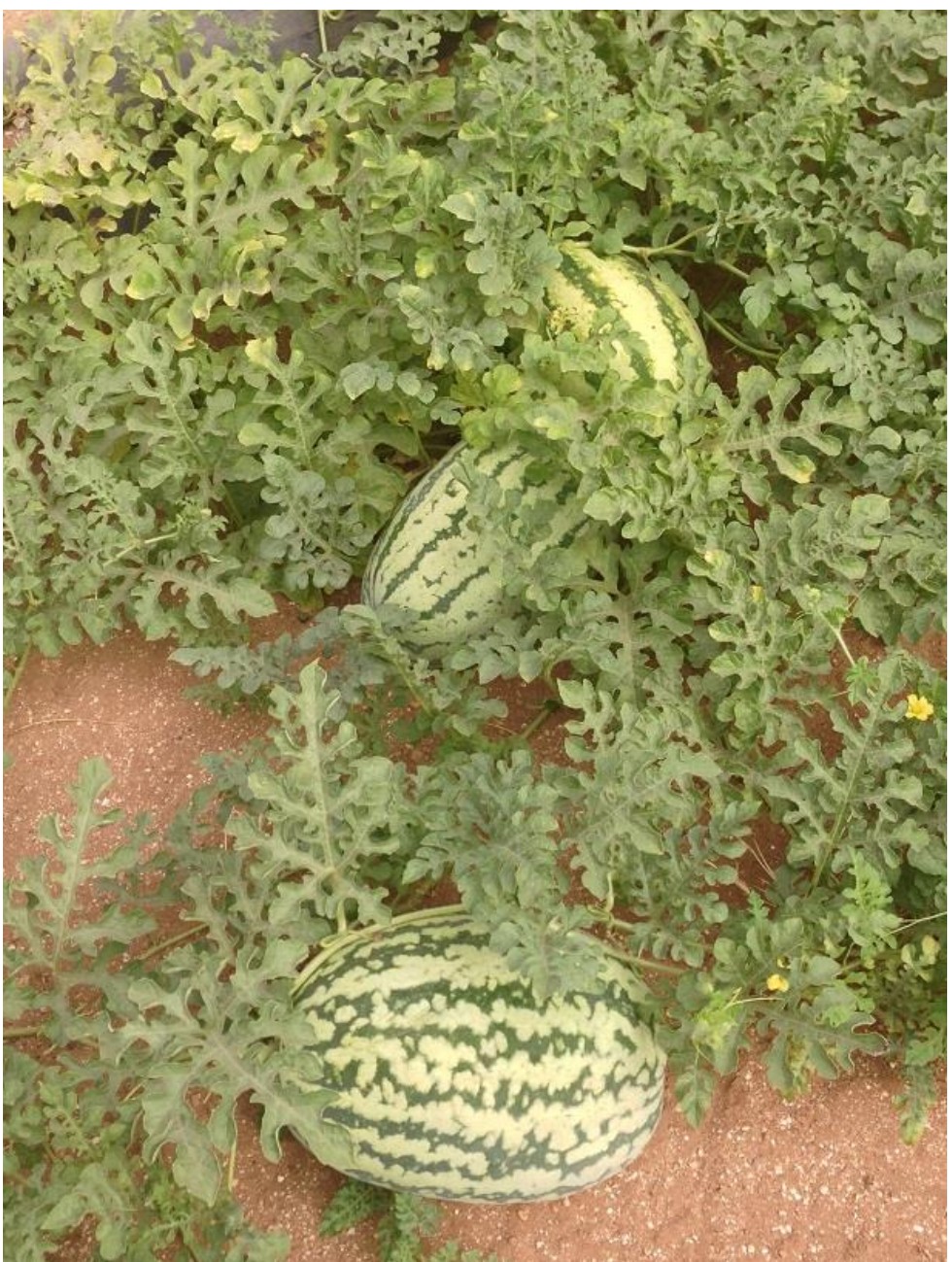

**Figure 4.** 'AitOulyad' watermelon landrace a few weeks prior to the first harvest. Notice the slight diversity in the striping pattern on the fruit.

### 3.2.5. Seed Numbers per Watermelon

Seed numbers obtained in watermelon fruit are an important characteristic as it is desirable for saving seed to plant the following year, but less seed is more desirable for the marketplace, as a seedy watermelon can make the flesh difficult to eat. 'AitOulyad' produced the highest numbers of seeds per fruit, followed by 'Bush Sugar Baby' and 'Rasmouka', with 'Orangelo' and 'All Sweet' having the lowest number of seeds in each fruit. 'All Sweet' produces an attractive red flesh with few seeds (Figure 5). Based on the evaluation, 'All Sweet' and 'Orangeglo' would be easiest to consume since they have less seed in the edible flesh, while others, especially 'AitOulyad' would be difficult to eat with such a seedy flesh. However, 'AitOulyad' has probably been selected to have high seed numbers for seed saving purposes.

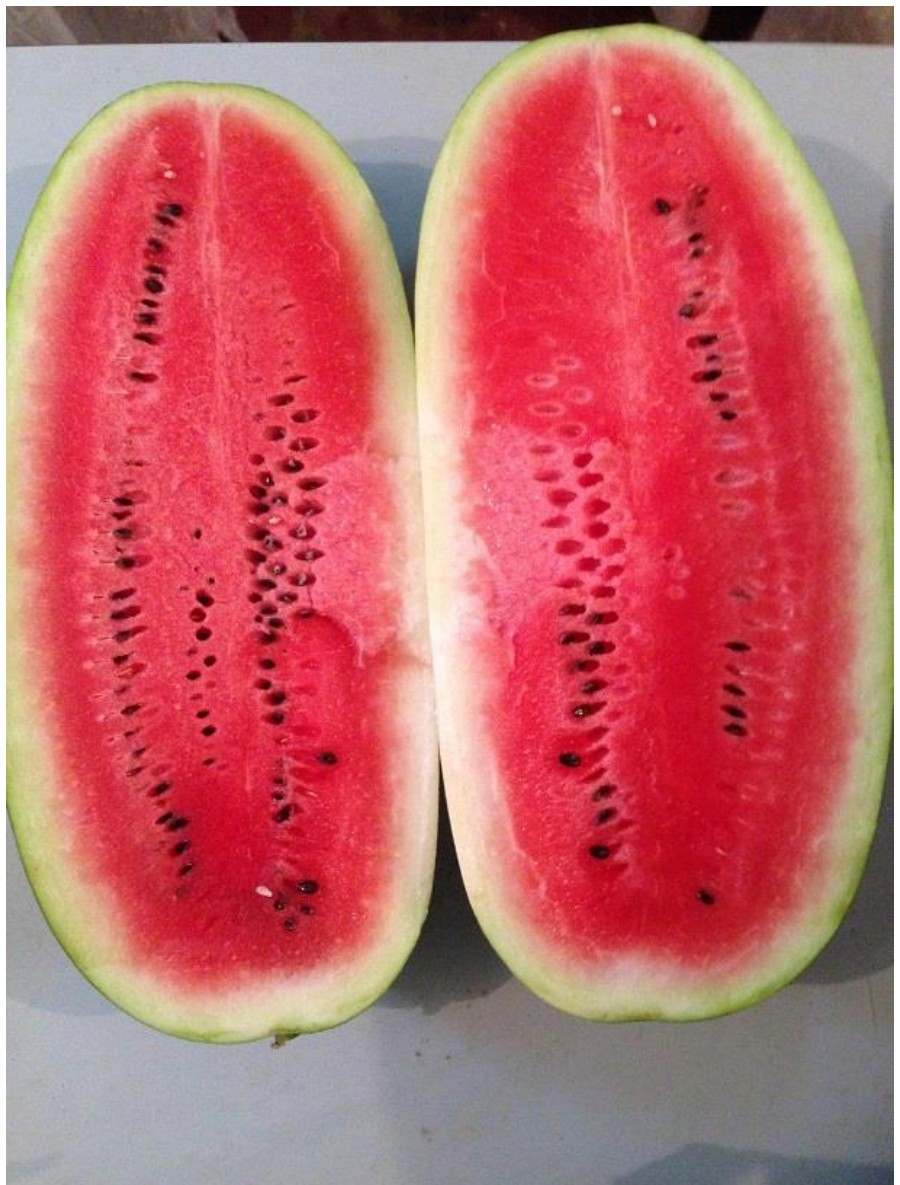

**Figure 5.** 'All Sweet' watermelon variety with attractive red flesh coloration and lack of seeds in flesh.

### 3.2.6. Summary for Watermelon Evaluation

This experiment revealed that watermelons landraces/pure line varieties differed for growth, disease susceptibility, and productivity. Plant vigor was generally highest in 'All Sweet' and 'Orangeglo' and was generally higher in those watermelon landraces/pure line varieties that were least affected by Fusarium wilt. Fruit shape and size consistency was 90% or greater, except for 'AitOulyad' at 80%. 'AitOulyad' produced both the highest fruit weights and numbers per ha, followed by 'All Sweet', 'Orangeglo', and 'Rasmouka'. 'All Sweet' and 'Orangeglo' both produced high average fruit weights followed by 'AitOulyad' and 'Rasmouka', with 'Bush Sugar Baby' being the lowest. For seed numbers per fruit, 'AitOulyad' produced the highest numbers of seed per fruit with 'All Sweet' having the least. These data indicate that 'AitOulyad' was the top landrace identified in this experiment, although 'Rasmouka' was comparable and better in some categories except for yield. However, considering growth, disease susceptibility, and productivity, 'All Sweet' was probably the best selection evaluated in this experiment and had excellent quality flesh with few seeds (Figure 5).

## 4. Discussion

Adaptation is an important factor that will shape the impacts of climate change on food production [14]. The high amount of genetic diversity in landrace crop populations allows them to adapt to drought, heat, saline soil, or other extreme environmental conditions which is essential for maintaining long-term productivity in stressful changing climates [1,2,7]. In developing countries having semi-arid to arid climates, melon and watermelon landraces are often used in cropping systems due to their low cost and adaptability to local growing conditions. In Morocco, landraces are still widely used for specific vegetable crops, especially in isolated areas that have traditionally maintained seeds for specific crops like melons and watermelons [7]. These Moroccan melon and watermelon landraces are an important source of genetic material to help cope and adapt to ongoing climate changes in the extreme growing conditions in southwestern Morocco.

Our results indicated that all melon landraces collected and evaluated are probably acceptable to help mitigate the influence of climate change in this area of Morocco, although some had better growth and productivity characteristics than others. 'AitOulyad' would be the top choice for a muskmelon due to its high plant vigor, yielding ability, average fruit size, and seeds produced per fruit. Although 'Ait Baha' muskmelon had soft flesh (rapid flesh breakdown) and did not have a long shelf life, this was a desired characteristic for some consumers. 'Casablanca' muskmelon produced a nice small melon having moderate disease development and low plant vigor, with low yields. 'Rasmouka Ananas' was an outstanding landrace with low foliar disease incidence, high plant vigor, high yields, high average fruit weight, and moderate seed production per fruit. 'Marrakech Souihla' had low amounts of foliar disease but produced low yields, poor fruit size and shape consistency, low average fruit size, and low numbers of seeds per fruit. All melon landraces apparently have adapted well to the current local growing conditions, although 'Casablanca' may have high yields under slightly cooler conditions, due to it being grown at a more coastal site in central Morocco. 'AitOulyad' muskmelon and 'Rasmouka Ananas' had the highest yields of all melon landraces evaluated, which is a direct result of human selection pressure over time under local conditions to improve their yielding ability. Melon landraces are known to exhibit great variation in fruit traits such as size, shape, color, taste, texture, and internal biochemical composition [15]. This variation results from the high genetic diversity within the landrace population that allows adaption to maintain high yields in the environmental extremes caused by climate change.

Similar to the melon study, watermelon landrace/pure line evaluation results indicated that some had better growth and productivity characteristic than others, but both landraces are probably acceptable to help mitigate the influence of climate change in this area of Morocco. 'All Sweet' and 'Orangeglo', two pure line varieties, had the outstanding potential to be grown in this environment in Southwestern Morocco (Table 2). Both had low foliar disease, moderate to high plant vigor, 90% fruit consistency, large fruit size, and low numbers of seeds per fruit. 'All Sweet' had high yield potential, while 'Orangeglo' had somewhat lower yields. The other pure line variety, 'Bush Sugar baby', had high disease susceptibility, low plant vigor, low yields, and low average fruit size. The two landraces were less desirable than the top pure line varieties, but 'AitOulyad' had high yields, which again is probably the result of human selection pressure causing these yield improvements in this environment. Watermelon landraces (such as those obtained from Southwestern Morocco and used herein) have often evolved under extreme dry and strenuous temperature climatic conditions. These traditional crop varieties have adapted well to the current changing environmental conditions to provide high yields [7]. This was observed specifically for the 'AitOulyad' watermelon landrace utilized in this study, which provided the highest yield of all landraces/pure lines evaluated.

The high morphological variation present in Moroccan watermelon landraces in this study are consistent with results from evaluations of local watermelon cultivars in North Africa, specifically Morocco [13] and Tunisia [16]. Moreover, these results along with ours

suggest the need to preserve this local germplasm for environmental adaptation to current and future climate changes.

Many dangers threaten the subsistence of landraces in Morocco, especially enhanced drought periods and increased temperatures via climate change, loss of traditional knowledge in relation to local genetic resources, traditional practices and uses, and the introduction of modern varieties [7]. However, landraces are being lost at an alarming rate in developing countries [9,10]. Moreover, Walters et. al. [7] indicated that a significant loss of vegetable crop landraces in the Souss-Massa region of Morocco has occurred during the last 30 years with approximately 80 to 90% of vegetable crop landraces lost during this time. The loss of vegetable landraces for use in small landowner production schemes has had a profound influence on hindering the amount of on-farm genetic perpetuation and conservation of this diverse and valuable genetic pool [17,18], which can be used to help mitigate the effect of climate change in fringe semi-arid to arid production regions [19]. Additionally, landraces are low-cost and sustainable, which is important to poor households in marginal environments that have limited financial resources available, such as those often found in southwestern Morocco [7].

Landraces provide diverse and dynamic gene pools that evolve over time under both farmer and natural selection pressure [9]. The high amounts of genetic diversity in landrace gene pools allow them to adapt to drought, heat, saline soil, or other extreme environmental conditions, which is essential to maintain long-term productivity especially during periods of erratic climate changes. Under changing climatic conditions, crop failures, yield reductions, quality loss, increasing insect and disease problems will become more prevalent and often have disastrous consequences for crop production [2,3], and landraces can help mitigate these problems with their high heterogeneity to allow evolutionary change to improve their productivity. Thus, landraces can be effectively used as a climate change adaptation strategy for small landholders [20].

## 5. Conclusions

The threat of a highly variable, unstable, and rapidly changing global climate is negatively impacting global crop production and compromising food security on a worldwide basis. The genetic diversity of crop landraces remains a crucial part of global food security, particularly for small landholders, subsistence agriculture in Africa, where increasing temperature and erratic precipitation are major threats due to rapid climate change. Agriculture is strongly influenced by climatic factors, with subsistence agriculture particularly vulnerable since small landholders do not have adequate financial resources to help adapt to climate change. In developing countries with arid to semi-arid climates, melon and watermelon landraces can be used to help mitigate the impact of climate change in regions with stressful environments. Our study indicated that all melon and watermelon landraces collected and evaluated in southwestern Morocco were acceptable to help cope with the impacts of climate change in this drought-prone area of North Africa, although some had better growth and productivity characteristics than others. The ability of small landholders to collectively conserve climate-adapted landraces can help mitigate future challenges of climate change. Moreover, the high genetic diversity in melon and watermelon landrace populations is important, as it allows these crop populations to adapt and help lessen the influence of climate change in these stressful environments. Growers must be willing to perpetuate these landraces through a selection of those that appear to be most adapted to climate change in future years. This will mitigate the influences of climate change over time. Future work should focus on continuing field evaluations of melon and watermelon landraces in arid and semi-arid regions of the developing world to determine ongoing climate change adaptability in these crop populations to ascertain those most suited to specific areas. Thus, landraces will continue to play a major role in maintaining world food security under a rapidly changing climate.

**Author Contributions:** S.A.W. developed the ideas for the project, analyzed data, interpreted results, and wrote most of the manuscript. R.B. and M.A. both helped to implement the study design and

edit the manuscript. R.B. also spent considerable time in project implementation and data collection. All authors have read and agreed to the published version of the manuscript.

**Funding:** U.S. State Department through Fulbright Scholar Program in Morocco (Moroccan-American Commission for Educational and Cultural Exchange).

**Institutional Review Board Statement:** Not required.

**Informed Consent Statement:** Not required.

**Acknowledgments:** This work was supported by the US State Department Fulbright Research Scholar Program (through the Moroccan-American Commission for Educational and Cultural Exchange), Southern Illinois University-Carbondale, USA, and the Moroccan National Institute of Agronomic Research, who all provided support for this project.

**Conflicts of Interest:** The authors declare no conflict of interest.

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
