# Peer review of "Local Melon and Watermelon Crop Populations to Moderate Yield Responses to Climate Change in North Africa"

_climate, doi:10.3390/cli9080129_

Round 1

Reviewer 1 Report

Dear Authors, 

I have gone through the paper entitled "Local Melon and Watermelon Crop Populations to Moderate Yield Responses to Climate Change in North Africa" an found that paper is written in good manor having sufficient novelties to be considered for publication. Further, melon and watermelon landraces in this area of southwestern Morocco with a semi-arid to arid climate will continue to play a major role in crop adaptation under a rapidly changing environment. Introduction has been written in support of state of the art of the study, methodologies used is fine, result and discussion is sound. Therefore, publication need minor corrections before accepted for publication.

Comment 1: Line no; 400-403 need support (references)

Comment 2: Conclusion need to be revised and re-written, kindly avoid the reference support in this section, this section is the key finding of the study, not required any support. 

Comment 3: Kindly make the table as per the journal guideline (fount is mis-matched) 

Author Response

Minor corrections:

Comment 1: I have included support references as requested

Comment 2: Conclusion was slightly modified and re-written to remove references, with more added regarding future direction

Comment 3: reworked the tables per journal guideline and corrected font

Reviewer 2 Report

Dear authors,

I consider that it is necessary to introduce a separate section of literature review and present previous work in the field, that means your references section must be improved. I also suggest that you follow the formatting rules of the journal.

The conclusion section must be improved, the managerial implications must be presented and future work.

Good luck! 

Author Response

I included another section in literature review as requested, see lines 71-89 and 91-103, also added additional references.

I went back through manuscript to correct formatting errors as requested.

I tried to improve the Conclusion section as requested, by describing both managerial implications and future work, see lines 442-449.

I incorporated your suggestions and I think it is a better manuscript now.